# Investigation of the Microstructure, Optical, Electrical and Nanomechanical Properties of ZnOx Thin Films Deposited by Magnetron Sputtering

**DOI:** 10.3390/ma15196551

**Published:** 2022-09-21

**Authors:** Michał Mazur, Agata Obstarczyk, Witold Posadowski, Jarosław Domaradzki, Szymon Kiełczawa, Artur Wiatrowski, Damian Wojcieszak, Małgorzata Kalisz, Marcin Grobelny, Jan Szmidt

**Affiliations:** 1Faculty of Electronics, Photonics and Microsystems, Wroclaw University of Science and Technology, Janiszewskiego 11/17, 50-372 Wroclaw, Poland; 2Faculty of Engineering and Economics, Ignacy Mościcki University of Applied Sciences in Ciechanów, Narutowicza 9, 06-400 Ciechanów, Poland; 3Faculty of Technical and Social Sciences, Ignacy Mościcki University of Applied Sciences in Ciechanów, Warszawska 52, 06-500 Mława, Poland; 4Institute of Microelectronics and Optoelectronics, Warsaw University of Technology, Koszykowa 75, 00-662 Warsaw, Poland

**Keywords:** ZnO thin film, pulsed reactive magnetron sputtering, hardness, scratch resistance, optical properties, electrical properties, structural properties

## Abstract

The paper presents the results of an investigation of the influence of technological parameters on the microstructure, optical, electrical and nanomechanical properties of zinc oxide coatings prepared using the pulsed reactive magnetron sputtering method. Three sets of ZnOx thin films were deposited in metallic, shallow dielectric and deep dielectric sputtering modes. Structural investigations showed that thin films deposited in the metallic mode were nanocrystalline with mixed hexagonal phases of metallic zinc and zinc oxide with crystallite size of 9.1 and 6.0 nm, respectively. On the contrary, the coatings deposited in both dielectric modes had a nanocrystalline ZnO structure with an average crystallite size smaller than 10 nm. Moreover, coatings deposited in the dielectric modes had an average transmission of 84% in the visible wavelength range, while thin films deposited in the metallic mode were opaque. Measurements of electrical properties revealed that the resistivity of as-deposited thin films was in the range of 10^−^^4^ Ωcm to 10^8^ Ωcm. Coatings deposited in the metallic mode had the lowest hardness of 2.2 GPa and the worst scratch resistance among all sputtered coatings, whereas the best mechanical properties were obtained for the film sputtered in the deep dielectric mode. The obtained hardness of 11.5 GPa is one of the highest reported to date in the literature for undoped ZnO.

## 1. Introduction

Zinc oxide (ZnO) and its compounds are now among the materials widely used in microelectronics. Due to the depletion of indium resources, transparent ZnO is an attractive semiconducting oxide that, with appropriate modifications, is a promising candidate for replacing indium-tin oxide thin films, commonly used in microelectronics for the fabrication of, among other things, transparent electrodes [1]. Zinc oxide is a wide band-gap semiconductor of the II-VI semiconductor group that has several favorable properties, such as good transparency in the wide wavelength range, high electron mobility, wide and direct band gap (3.37 eV), high exciton binding energy (60 meV) and strong room temperature luminescence [2,3,4,5]. Moreover, ZnO is generally an n-type transparent oxide semiconducting material [4]. However, an additional advantage is the possibility of the preparation of p-type transparent conductive coatings using in situ doping of ZnO during its deposition [6]. The form type and dimensions of ZnO nanostructures have a strong impact on its features as well [7,8,9]. All of the listed properties and additionally the high thermal and mechanical stability at room temperature make it attractive for potential use in electronics, optoelectronics and laser technology [10]. ZnO has found applications in the manufacturing of devices such as varistors, gas sensors, transparent electrodes for thin-film transistors and solar cells. The other emerging applications include light emitting diodes (LEDs), laser diodes (LDs) and light detectors [1,3,11]. ZnO is often used for the preparation of various junctions and heterostructures [12,13,14,15]. The piezoelectric and pyroelectric properties of ZnO make it possible to use in sensors, converters, energy generators and photocatalysts in hydrogen production [11,16]. Due to its hardness, stiffness and piezoelectric constant, it is an important material in the ceramics industry, while its low toxicity, biocompatibility and biodegradability result in increased interest for its application in biomedicine and in ecological systems [10,17].

ZnO thin films can be prepared on an industrial scale using many methods [3,10,18,19]. One of them is magnetron sputtering, which ensures a stable and high-efficiency deposition process and provides the possibility of obtaining coatings on large-scale substrates. Moreover, in this method, the properties of the ZnO film can be controlled by changing the sputtering conditions such as substrate temperature, deposition time, pressure and power [3]. There are many types of magnetron sputtering methods, including direct current (DC), radio frequency (RF) and high-power impulse magnetron sputtering (HiPIMS), among others. Radio frequency magnetron sputtering RF MS (13.56 MHz, 27 MHz) allows the sputtering of nonconductive and conductive materials. In the case of the deposition of non-conducting ZnO, a metallic Zn target is usually sputtered in an oxygen atmosphere. In each case, partial pressure is an important process parameter, and usually the target power density does not exceed 10 W/cm^2^ [2,3,20,21,22,23,24,25,26,27,28,29,30,31,32]. Controlling the level of oxygen doping during the RF process allows one to control the stoichiometry of the deposited films. It is also possible to obtain Zn oxide coatings using DC MS by sputtering the conductive target. In most cases, this forces the use of targets made of metallic Zn material. As in the case of RF MS, the target power density also does not exceed 10 W/cm^2^ [33,34,35,36,37,38,39,40,41,42,43]. Recently, a lot of attention has been focused on high-power impulse magnetron sputtering [44,45,46]. In this method, the target is sputtered with very high and short current pulses in which duration is on the order of a dozen or several dozen microseconds. Current waveforms are modulated at a frequency of several kHz. However, it is worth highlighting that the average power released in the target is similar to standard DC and RF MS processes (although the sputtering efficiency is relatively smaller). In turn, with the HiPIMS method, mechanical properties such as hardness or optical properties of deposited films can be enhanced due to phenomena accompanying the initiation and pulse suppression.

In the present paper, the influence of the oxygen content in the sputtering atmosphere on various properties of zinc oxide thin films was discussed. It was found that the change in the sputtering mode had a significant effect on the microstructure, morphology, electrical, optical and mechanical properties of the sputtered ZnOx thin films. It is also worth noting that the hardness determined for the ZnO thin film deposited in the deep dielectric mode has been one of the highest reported in the literature and the coating had excellent scratch resistance.

## 2. Materials and Methods

For the deposition of ZnOx, thin films’ reactive pulsed magnetron sputtering process of metallic zinc targets was used. Deposition processes were carried out in an NP-500 vacuum stand setup (Bolesławiec, Poland) equipped with a pumping system (Tepro, Koszalin, Poland) consisting of diffusion (2000 L/s) and rotary (30 m^3^/h) pumps. The final pressure obtained with a vacuum set was equal to 2∙× 10^-5^ Torr. Each sputtering process was carried out in the atmosphere containing a mixture of argon and oxygen gases (Ar:O_2_), and the working pressure of the gas atmosphere was changed in the range from 1 × 10^−3^ to 8 × 10^−3^ Torr. Circular WMK-100 magnetron (handmade) was used, which was equipped with a high purity circular Zn (99.999%) disc with a diameter of 100 mm and a thickness of 9 mm. The used magnetron offers the possibility of conducting processes with a very high power released in the sputtered target, which is directly related to the efficiency of the deposition processes of thin films. The magnetron was powered using an MSS-10kW power supply unit (Dora Power System—PWM type, Wilczyce, Poland). A distinctive feature of this power supply unit is the possibility of in situ observations of sputtering processes by monitoring its parameters [47]. In particular, the most important parameter is called the circulating power (the term introduced by the Dora Power System manufacturer), reflecting changes in a load impedance. In turn, impedance is determined by specific technological conditions of the sputtering process and is affected, for example, by the change of the: (1) type and composition of the working gas, (2) target thickness, and (3) surface of the sputtered material (e.g., coverage of the target with a reactive compound formed in the presence of chemically active gases). This is an original feature of the DPS power supply that can be used to control the deposition process. Thin films were deposited on fused silica (SiO_2_), silicon (Si) and alumina substrates containing interdigitated electrodes designed for electrical measurements. The thickness of the deposited ZnOx thin films was assessed with the use of the Taylor-Hobson CCI Lite optical profilometer (Leicester, UK).

The microstructure was studied by X-ray diffraction in grazing incidence mode (GIXRD) using a Panalytical X’Pert Pro diffractometer (Panalytical, Malvern, UK) equipped with the Cu Kα X-ray source with a wavelength of 1.5406 Å. Crystallite size analysis was performed according to the Debye-Scherrer equation [48] with the aid of MDI JADE 5.0 software. The surface morphology and cross section of as-prepared coatings were investigated using a FEI Helios Xe-PFIB field-emission scanning electron microscope (FE-SEM) equipped with EDAX energy dispersive spectroscopy (EDS) detector (Hillsboro, OR, USA).

Optical measurements were conducted in the wavelength range of 300–1000 nm using an Ocean Optics QE65000 spectrophotometer and a coupled deuterium-halogen light source (Ocean Optics, Largo, FL, USA). The measurements of transparent ZnOx thin films were completed by evaluating the fundamental absorption edge (λ_cut-off_). The refractive index (n) and extinction coefficient (k) spectral characteristics of the transparent films were determined by the reverse engineering method using the FTG FilmStar software (Princeton, NJ, USA) employing a generalized Cauchy model for materials with k > 0. 

The resistivity of the deposited coatings was determined using current-voltage characteristics measured at room temperature in a dark shielded enclosure with the aid of a Keithley SCS4200 system (Keithley Instruments LLC, Cleveland, OH, USA) and a M100 Cascade Microtech probe station (Cascade Microtech, Beaverton, OR, USA).

The mechanical properties of prepared zinc oxide thin films were determined using a nanoindentation method with the aid of a CSM Instruments (Peseux, Switzerland) nanoindenter equipped with a diamond Vickers tip. The results were analyzed using the method proposed by Oliver and Pharr [49]. In turn, the scratch resistance of the deposited zinc oxide thin films was measured using the Bayer test and the Taber Oscillating Abrasion tester 6160 (North Towananda, NY, USA). The oscillating movement of the abrasive medium, that is, 6/9 silica sand, simulates the everyday wear of the measured coating. The resistance to abrasion was determined using the oscillating sand test described in ASTM F735 [50]. The movement distance of the sand tray was equal to 100 mm at a speed of 300 strokes per minute. The surface topography of the deposited thin films was examined before and after the scratch test using a TalySurf CCI Lite Taylor-Hobson optical profiler and an Olympus BX51 optical microscope (Bartlett, TN, USA). As an indicator of scratch resistance (hereinafter referred to as *SRes*), a comparison was made of the mean square root height (*Sq*) measured before and after the abrasion test as in Equation (1):(1)$$SRes=\frac{Sq_{before}}{Sq_{after}}\times 100\%$$
where *Sq_before_* is the mean square root height before abrasion test and *Sq_after_* is the mean square root height after abrasion test.

It can be estimated that the closer the *SRes* is to 100%, the more resistant the coating is to scratching due to the negligible change in the roughness of its surface.

## 3. Results and Discussion

### 3.1. Deposition of ZnOx Thin Films

Preliminary experiments performed by the authors have shown that the relation between the effective power P_E_ (proportional to thin film deposition rate) and circulating power P_C_ (as an indicator of the target surface condition) of the power supply (P_E_/P_C_) measured during sputtering in the argon atmosphere allows us to calibrate and scale magnetron sputtering processes in the atmosphere of the argon and oxygen mixture. The sputtering of the Zn target in an Ar atmosphere with effective power of P_E_ = 1 kW (target power density of ca. 12 W/cm^2^) and substrate target distance (d_S-T_) equal to 130 mm resulted in the Zn film deposition rate equal to 300 nm/min. In turn, P_E_ equal to 6.1 kW (target power density of ca. 78 W/cm^2^) resulted in a deposition rate of Zn films of ca. 2000 nm/min. Additionally, changing the d_S-T_ to 90 mm and providing the P_E_ of 1.6 kW (target power density of approximately 20 W/cm^2^) resulted in a decrease in the deposition rate to approximately 800 nm/min.

In this paper, ZnOx thin films were deposited by sputtering a Zn target in pure oxygen and also in the atmosphere of the argon and oxygen mixture. In each process, the composition of the sputtering atmosphere and power supplying target (process parameters) were changed. The dependence of the effective power on the circulating power P_E_/P_C_ (Figure 1) measured during sputtering in pure argon and pure oxygen allowed us to determine the boundary electrical parameters of the power supply when the target surface was still metallic (metallic mode) or it was already fully oxidized (dielectric mode). The deposition of thin films using reactive magnetron sputtering required the determination of the most favorable technological conditions of the process. The most important requirements are the composition of the working gas and the total pressure of the gas mixture during the sputtering process. The nature of the reactive sputtering process forced the determination of certain procedures while setting the gas level, which was possible due to the in situ observation of the sputtering process based on the value of circulating power.

Observation of the circulating power gives information on the surface condition of the target during reactive sputtering. Covering a sputtered target with dielectric compound, i.e., its oxidation, changes the electrical conditions of its surface because the secondary electron emission factors from the dielectric layer are, on average, one order larger than those for a pure metal. The relationship between the DPS power supply (effective power versus circulating power) and the surface of the sputtered material mirrors the target surface status (Figure 2).

In particular, the P_E_/P_C_ value is an important parameter of the reactive sputtering process, providing information about the target surface coverage (depending on the magnetron sputtering mode during the deposition process). The formation of a reactive compound on the metallic target changes the physicochemical properties of the surface, leading to a significant reduction in the depFosition rate. The sputtering yield of a dielectric compound is often an order of magnitude lower than that of metals. The increase in the circulating power value always clearly indicates the technological point when the deposited dielectric thin films are transparent, hard and exhibit good adhesion to the substrate.

The hysteresis effect of the reactive magnetron sputtering process is the result of phenomena occurring on the surface of the sputtered material. The metallic target is an efficient source of sputtered metal (zinc) vapor. When the target surface is covered with zinc oxide, the sputtering efficiency significantly decreases. As a result, there is ambiguity in the sputtering conditions of the target, whose surface may be unstable (balance between the formation of the compound and its etching from the target). This is due to the hysteresis phenomenon that occurs if the sputtering parameters, such as target power and reactive gas flow, are changed. Figure 3 presents the hysteresis phenomenon of the dependence of P_C_ = f(O_2_) for sputtering processes with an effective power equal to 1.6 kW. It can be concluded that the oxide layer on the target starts to grow at p_O2_ = ~5 × 10^−3^ Torr, and the point (*c*2) describes the conditions under which the Zn target is covered by the dielectric ZnOx compound. The reduction of the partial pressure of oxygen does not cause an immediate return of the magnetron to the metallic mode but the gradual shift of the work point to the (*c*1) point, while the target is still covered with oxide. Decreasing the partial pressure of oxygen below p_O2_ = ~3∙× 10^-3^ Torr causes a change from oxide to metallic mode, which means that the Zn target is no longer poisoned with an oxide ((*a*) point). The authors named points (*c*1) and (*c*2) as the shallow dielectric mode and the deep dielectric mode, respectively. With the dynamically determined partial pressure of oxygen (based on the circulating power value) at a constant argon pressure, the characteristics of P_C_ = f(p_O2_) were measured for an effective power equal to 1.6 kW. As can be seen in Figure 3, the hysteresis effect was observed in the entire effective power range.

The sputtering efficiency of ZnOx films was measured for coatings deposited at an effective power of 1.6 kW (Figure 4). As the oxygen content in the mixed Ar and O_2_ atmosphere increased, from a certain percentage of oxygen in the Ar:O_2_ atmosphere, a rapid decrease in the deposition rate of thin films was observed with a simultaneous rapid increase in the circulating power. Under these technological conditions, the Zn target was completely oxidized. 

Conductive and opaque thin films were obtained despite the fact that all sputtering processes were carried out at a relatively high concentration of O_2_ in the Ar:O_2_ atmosphere. From a certain oxygen concentration in the sputtering atmosphere, the deposited films were transparent and exhibited good adhesion to the substrate. The shaded area marked in Figure 4 identifies a transparent zinc oxide film with good adhesion to the substrate. At the set target power level, the oxide coatings were sputtered with a deposition rate of ~20 nm/min. The gradual increase of the oxygen content in the sputtering process caused an increase in the resistivity of the deposited films (Figure 4) up to the point where they became transparent and electrically nonconductive. It was observed that when the O_2_ content was greater than 85% in the Ar:O_2_ atmosphere, thin films became dielectric. The percentage content of oxygen in the working atmosphere was estimated on the partial pressure basis of the values measured after the sputtering process. The determination of the limit of oxygen content in the sputtering process, simultaneously defining the metallic and dielectric modes, was consistent with the value of the circulating power during the sputtering process.

In this work, three sets of ZnOx thin films were deposited with different technological parameters. ZnOx coatings were prepared with various and dynamically determined partial pressure of oxygen. The Zn target was sputtered in three different modes: (*a*) metallic, (*c*1) shallow dielectric and (*c*2) deep dielectric (Figure 3). Moreover, the deposition time was also matched with the sputtering modes and was equal to 30, 300 and 480 s for metallic, shallow and deep dielectric modes, respectively. Different sputtering times were related to the deposition rate and were selected so that the thickness of the films was in the range of 500 nm. The thickness of the films deposited in the metallic mode (*a*) was equal to 450 nm, whereas the thickness of the coatings deposited in the shallow (*c*1) and deep dielectric (*c*2) modes was equal to 350 nm and 570 nm, respectively.

### 3.2. Structural Properties of ZnOx Thin Films

XRD analysis was performed to determine the structural properties of the prepared ZnOx thin films (Figure 5). It was found that the diffraction pattern obtained for coatings deposited at the (*a*) work point (metallic mode) revealed the coexistence of metallic and oxide hexagonal phases of Zn and ZnO. The deposited thin film was nanocrystalline with small crystallites with size equal to 9.1 and 6.0 nm for Zn and ZnO, respectively. Furthermore, the zinc oxide coatings deposited at working points (*c*1) and (*c*2) had the polycrystalline structure of the hexagonal ZnO phase with the most dominant peak occurring at 2θ = 34° and associated with the (002) lattice plane. The lattice planes observed for the ZnO film deposited in the deep dielectric mode (*c*2 work point) were more intense than for the film deposited in the shallow dielectric mode (*c*1 point). In the case of ZnO films deposited at the working points (*c*1) and (*c*2), the crystallite size was equal to 7.4 nm and 6.7 nm, respectively. XRD measurements performed for thin films sputtered in dielectric modes revealed a shift of the diffraction peaks related to the ZnO hexagonal phase towards a lower angle (2θ), which may indicate the presence of a tensile stress occurring in the structure. In turn, in the case of coatings sputtered in the metallic mode, there was a visible shift of the diffraction peaks related to the metallic Zn-hexagonal phase toward a higher angle, which can testify about the presence of a compression stress. The results of the XRD measurements are summarized in Table 1.

The surface and cross-section morphology of the deposited thin films at different partial pressure of oxygen p_O2_ was obtained using FE-SEM and shown in Figure 6 with thickness markers at the cross sections. It was found that the surface of the ZnOx coatings sputtered in the metallic mode was composed of spherical-shaped nanosized grains with various sizes and slight voids between them. The cross section showed that the microstructure consisted of coarse grains and that the thin film was very rough. In turn, coatings deposited in shallow and deep dielectric modes had a very smooth surface composed of homogeneously distributed nanograins with almost uniform size. The cross-sectional morphology revealed columnar growth of the coatings and the fact that a change in the sputtering mode caused densification of the thin films. Furthermore, the columnar growth started from the substrate, and the columns tended to thicken their diameter during crystal growth.

Secondary electron images, elemental distribution maps of zinc and oxygen and EDS spectra taken for ZnOx thin films deposited in metallic and deep dielectric modes are shown in Figure 7. The gathered maps (Figure 7a) showed homogenous distribution of each element, i.e., Zn and O. EDS spectra (Figure 7b) showed peak lines related to Zn (Lα line at 1.01 keV) and O (Kα line at 0.53 keV) from the thin film and Si from substrate material (Kα line at 1.74 keV). It is clearly seen that for the deep dielectric mode, the oxygen-related peak is much higher compared to the thin film from the metallic mode, which is understandable given the nature of both modes of the magnetron sputtering process. Furthermore, there were no other peaks that could come from other elements, which is evidence of the good purity of the deposited zinc oxide thin films.

### 3.3. Optical Properties of ZnOx Thin Films

The optical properties of the deposited ZnOx thin films were determined based on measurements of light transmission in the wavelength range of 300–1000 nm (Figure 8a). It was found that coatings sputtered in the metallic mode were opaque in the measured wavelength range, although their microstructure consisted of both metallic Zn and oxide ZnO phases. In turn, the ZnOx films deposited in the shallow dielectric (*c*1) and deep dielectric (*c*2) modes were well transparent with an average transparency of approximately 84%. The results of the determination of the position of the fundamental absorption edge are also presented in Figure 8a. The lowest cut-off wavelength (λ_cut-off_) was equal to 361 nm for the ZnOx film deposited in the shallow dielectric mode (*c*1 work point). The change in the deposition mode to (*c*2) work point resulted in a slight shift of the fundamental absorption edge towards a longer wavelength of 370 nm. Moreover, the optical band gap energy (E_g_^opt^) of the ZnOx thin films was calculated using the Tauc plot [51]. The optical band gap energy was estimated by extrapolating the linear portion of the curves presented in Figure 8b. In the case of ZnO films deposited in shallow dielectric and deep dielectric modes, the value of E_g_^opt^ was equal to 3.30 eV and 3.25 eV, respectively.

Based on the results of the transmission characteristics (Figure 8a), with the aid of the reverse engineering method and Film Star software (v. 2.61), the refractive index (n) and the extinction coefficient (k) were calculated and plotted in Figure 9. It was found that the n at λ = 550 nm for nanocrystalline ZnOx thin films deposited in shallow and deep dielectric modes were approximately 1.97–1.98 (Figure 9a). Furthermore, the analysis of the extinction coefficient showed that the optical losses are two times higher for the ZnOx film deposited in the shallow dielectric mode for which the extinction coefficient was equal to 6.2 × 10^−3^ (Figure 9b). However, the obtained values of the extinction coefficient testify about very good optical properties of both coatings.

### 3.4. Electrical Properties of ZnOx Thin Films

Based on the current-voltage characteristics measured in a planar configuration, the values of sheet resistance (R_sheet_) and resistivity (ρ) were determined. The resistivity of the nanocrystalline thin film deposited in metallic mode was equal to 3.8 × 10^−4^ Ωcm, which clearly showed that the occurrence of the oxide ZnO-hexagonal phase in its structure did not cause a significant increase of ρ as compared to pure metallic Zn. For the prepared ZnO coatings in dielectric modes (*c*1; *c*2), the resistivity was equal to 3.2 × 10^7^ Ωcm and 4.5 × 10^8^ Ωcm, respectively. The current-voltage characteristics for ZnOx thin films are presented in Figure 10. Based on these results, one can observe that different technological parameters of the pulsed reactive magnetron sputtering process had considerable influence on electrical parameters. The ZnOx thin films deposited with different partial pressure of oxygen p_O2_ at the (*a*) work point were significantly different compared to the films deposited at (*c*1) and (*c*2) work points.

### 3.5. Mechanical Properties of ZnOx Thin Films

The results of the nanoindentation investigations of ZnOx thin films are shown in Figure 11. The hardness of the deposited coatings varied significantly between films deposited in different sputtering modes. The lowest hardness was obtained for ZnOx thin film deposited in the metallic mode. This could be anticipated due to the low hardness of metallic zinc whose hardness on the Mohs scale is only 2.5; therefore, it is the same as rather soft silver and magnesium and slightly harder than lead, known as a soft metal. In the case of the thin films deposited in the dielectric modes, their hardness was much higher and equal to 8.5 and 11.5 GPa for the film sputtered in a shallow and deep dielectric mode, respectively. The first value is consistent with the literature reports, where the hardness of ZnO thin films is usually in the range of 5 to 8 GPa. However, values as low as 2 GPa or as high as 10.8 GPa can also be found [52,53,54,55,56,57,58,59,60,61,62]. It is worth noting that the hardness obtained for ZnOx thin films sputtered in the deep dielectric mode was one of the highest in all of the literature reports to date. This might be caused by the highly nanocrystalline structure, where the crystallite size is only 8.1 nm. In the case of the ZnOx thin films deposited in shallow dielectric mode, the crystallite size is slightly larger and equal to 8.7 nm, which might be the reason for the lower hardness due to the Hall–Petch effect. However, both thin films had a smooth surface with homogenously distributed nanograins and exhibited a densely packed, void-free, nonporous columnar microstructure. The refractive index of the film prepared in the deep dielectric mode is slightly higher than that of the shallow dielectric mode. At λ = 500 nm, n is equal to 2.01 and 1.99 for deep and shallow dielectric modes, respectively. Both values are close to those of a single ZnO crystal, which is equal to 2.05 at the same wavelength [63]. Furthermore, both values are rather high compared to literature reports, where n is usually in the range of 1.9 to 2 [64,65,66,67,68]. On the basis of the values of the refractive index, the packing density of both thin films was determined according to the Equation (2) [69,70]:(2)$$P=\frac{\left(n_{f}{}^{2}-1\right)\cdot \left(n_{b}{}^{2}+1\right)}{\left(n_{f}{}^{2}+2\right)\cdot \left(n_{b}{}^{2}-1\right)}$$
where: *n_f_* is the refractive index of ZnO thin films at 500 nm and *n_b_* is the refractive index of bulk ZnO at 500 nm.

In this way, a direct comparison could be made between the deposited thin films. The better packing density was found for the ZnOx thin film prepared in the deep dielectric mode and was equal to 0.972 compared to 0.964 obtained for shallow dielectric mode conditions. These results are consistent with the hardness measurements and the crystallite size of both types of coatings. 

Furthermore, the abrasion resistance of each deposited thin film was estimated based on the Bayer test. This parameter is especially important in the case of protective transparent coatings that may be applied in various fields of optics or optoelectronics, e.g., in ophthalmics or touch screens. The investigation results are shown in Figure 12. Each thin film was measured before and after performing the scratch test using an optical microscope working in reflection mode and an optical profilometer to obtain the 3D images of the surface and 2D surface cross-sectional profiles. Based on these data, surface roughness was determined using the root mean square height (*Sq*), and its change was used to evaluate scratch resistance (*SRes*) (Equation (1)). Each sample had a homogeneous and crack-free surface before performing the scratch test, and the *Sq* value was in the range of 1.06 to 1.25 nm. After performing the Bayer test, the metallic coating was rough, containing visible scratch lines with a depth of even 400 nm. The *Sq* changed significantly and the *SRes* value was only 1.64%, which clearly shows that this coating is not scratch resistant. In turn, both thin films sputtered in dielectric modes looked similar before and after the test. The 3D image of the surface and surface cross-section profile for thin films from the shallow dielectric mode showed some minor changes, while the *Sq* value increased slightly. Moreover, the optical microscope observations also showed some very small and difficult-to-see scratches. This resulted in a *SRes* value of 78.9%, which indicates fairly good scratch resistance. However, the best ZnO thin films were deposited in the deep dielectric mode. The images of the optical microscope and profilometer did not show almost any change in the surface and *Sq* value after the Bayer test, resulting in a *SRes* of 96.9%. This testifies about a very good scratch resistance of these coatings.

Summary of the obtained results for ZnOx thin films is shown in Table 2.

## 4. Conclusions

The ZnOx thin films were deposited with different oxygen concentrations in the sputtering atmosphere, which determined three different sputtering modes: metallic, shallow dielectric and deep dielectric modes. All prepared ZnOx films were obtained with a relatively high deposition rate, which was related to the high target power density during the sputtering process, which was not found in the other works. Microstructure studies showed that thin films deposited in the metallic mode had mixed metallic (Zn) and oxide (ZnO) hexagonal crystal phases with crystallite size of 9.1 and 6.0 nm, respectively. Moreover, thin films deposited in shallow and deep dielectric modes had a polycrystalline structure of the hexagonal ZnO phase with crystallite size of 8.1 to 8.7 nm. It was also found that coatings deposited in the metallic mode were opaque, which was related to the occurrence of metallic Zn in the deposited thin film. On the contrary, thin films deposited at shallow and dielectric modes were highly transparent, with an average transparency in the visible wavelength range above 84%. On the basis of the extinction coefficient values, it can be concluded that the optical losses are almost two times smaller for ZnOx coatings deposited in the deep dielectric mode compared to films deposited in the shallow dielectric mode. Measurements of electrical properties revealed that the resistivity of as-deposited thin films was dependent on the sputtering mode. For coatings deposited in the metallic mode, the resistivity was equal to 3.8 × 10^−4^ Ωcm, while for films deposited in the dielectric modes, it was significantly higher and was in the range from 10^7^ to 10^8^ Ωcm. Measurements of nanomechanical properties revealed that metallic thin films were rather soft with hardness of 2.2 GPa, but it is worth mentioning that coatings from the deep dielectric mode had one of the highest values of hardness ever reported, i.e., 11.5 Gpa. Moreover, coatings deposited in the dielectric modes were scratch resistant, and the best abrasive properties were obtained for the ZnOx thin film prepared in deep dielectric mode.

## Figures and Tables

**Figure 1 materials-15-06551-f001:**
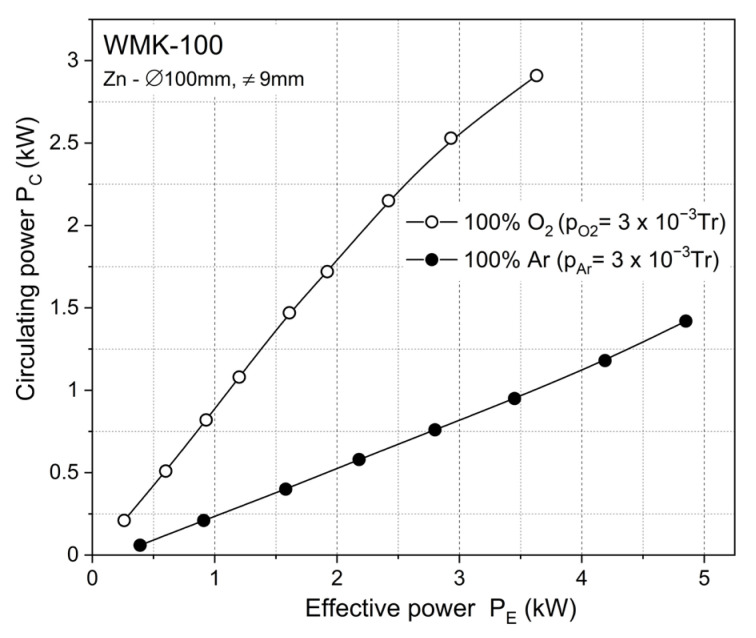
Dependence between effective power (P_E_) and circulating power (PC) measured during the sputtering of Zn target in Ar (pure metallic target surface—metallic mode) and O_2_ atmosphere (fully oxidized target surface—dielectric mode).

**Figure 2 materials-15-06551-f002:**
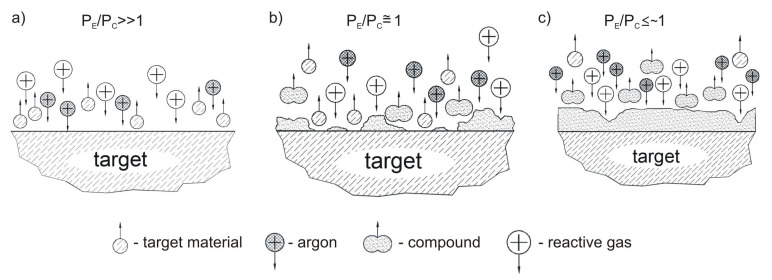
Schematic view of the formation of the reactive compound on the surface of the sputtered material as a function of the electrical parameters of the DPS power supply: (**a**) metallic mode (Ar atmosphere)—target surface not covered by the compound; (**b**) shallow dielectric mode—target surface partially covered by a compound; (**c**) deep dielectric mode (O_2_ atmosphere)—target surface completely coated with the compound.

**Figure 3 materials-15-06551-f003:**
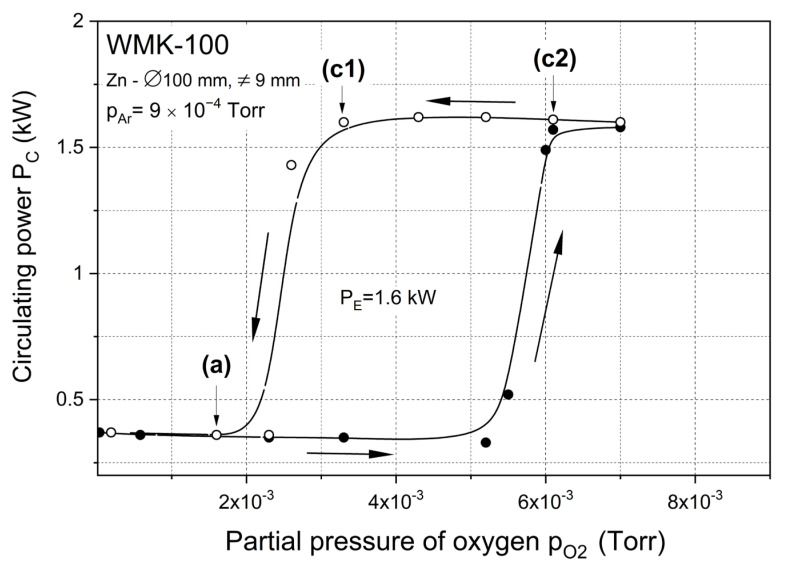
Influence of the partial pressure of oxygen on the value of the circulating power during the reactive sputtering of the Zn target in the mixed Ar and O_2_ atmosphere at P_E_ = 1.6 kW.

**Figure 4 materials-15-06551-f004:**
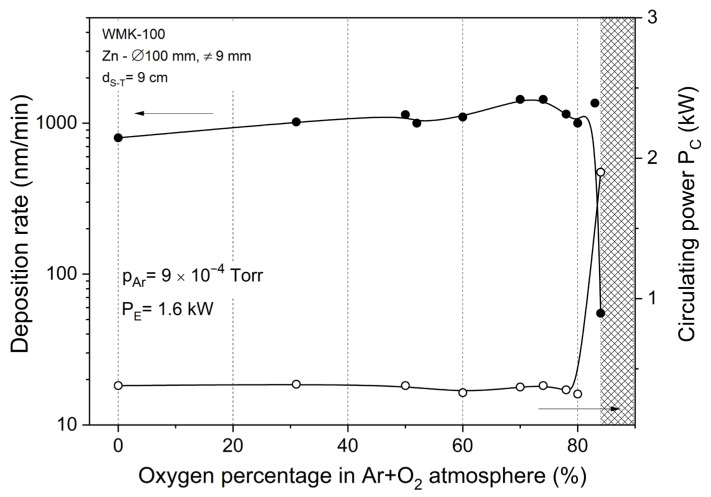
Influence of the percentage of oxygen content in the Ar:O_2_ atmosphere on the deposition rate of the ZnOx films and circulating power.

**Figure 5 materials-15-06551-f005:**
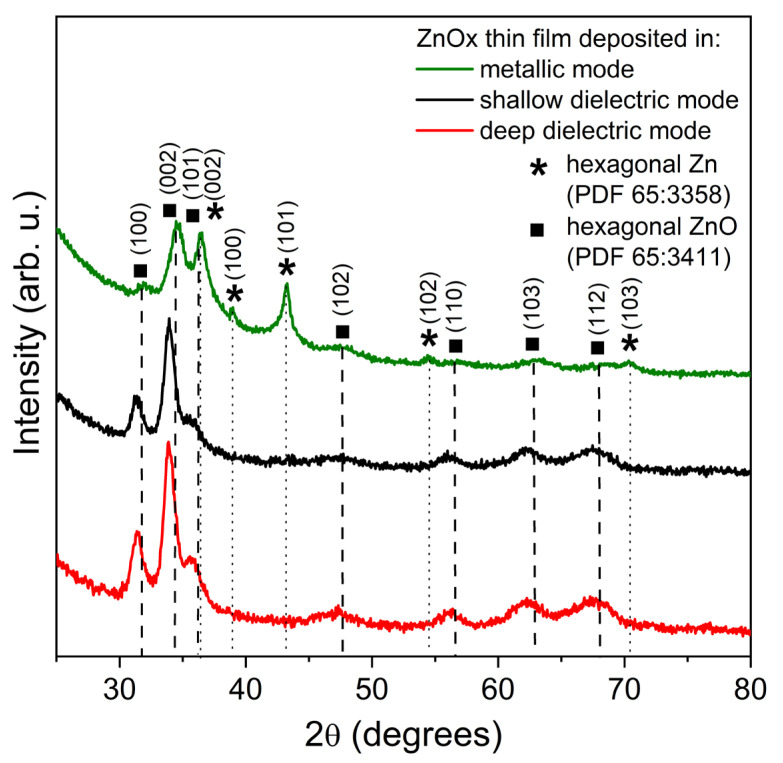
Diffraction patterns of ZnOx thin films deposited in metallic mode (a) work point, shallow dielectric mode (*c*1) work point and deep dielectric mode (*c*2) work point.

**Figure 6 materials-15-06551-f006:**
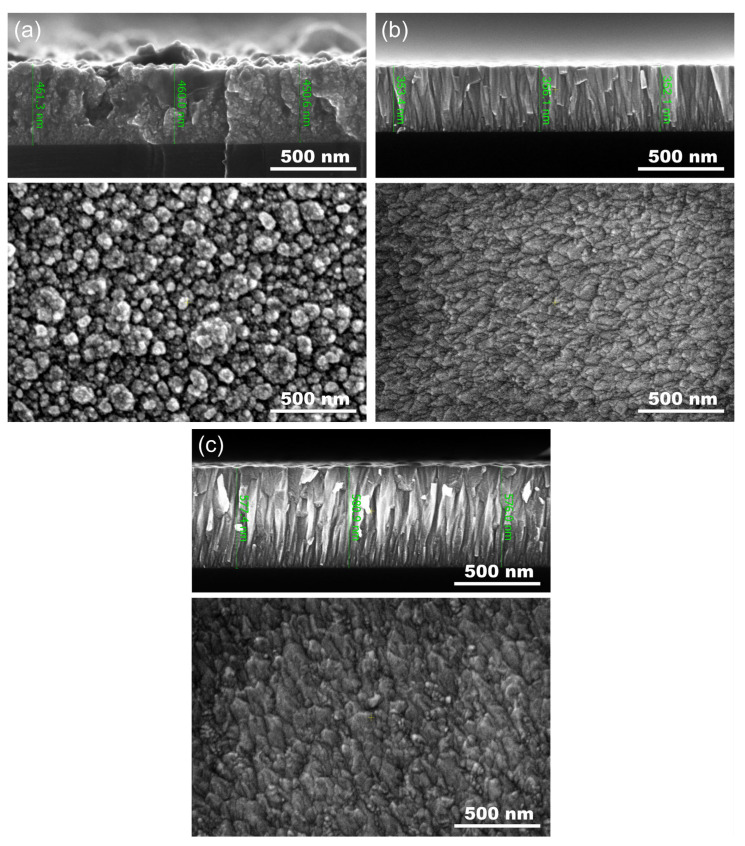
SEM images of ZnOx thin films deposited in modes: (**a**) metallic, (**b**) shallow dielectric and (**c**) deep dielectric.

**Figure 7 materials-15-06551-f007:**
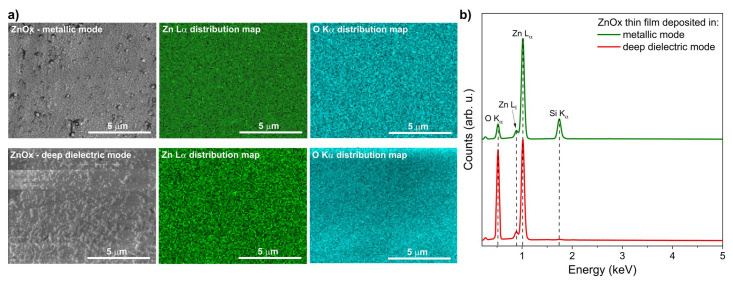
Results of material composition investigation: (**a**) elemental distribution maps and (**b**) EDS spectra of ZnOx thin films deposited in metallic and deep dielectric modes.

**Figure 8 materials-15-06551-f008:**
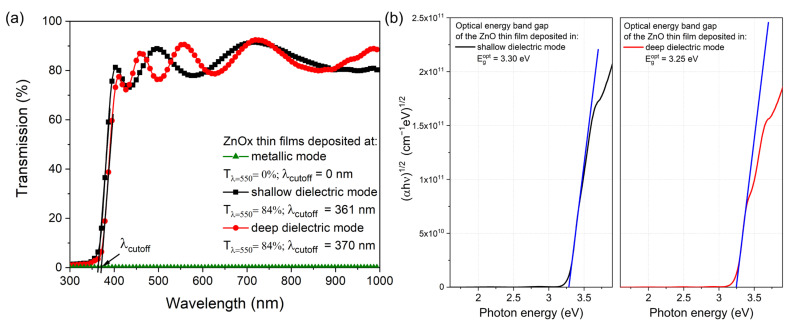
Transmission spectra (**a**) and Tauc plots (**b**) of transparent ZnOx thin films prepared in shallow dielectric and deep dielectric modes of magnetron sputtering.

**Figure 9 materials-15-06551-f009:**
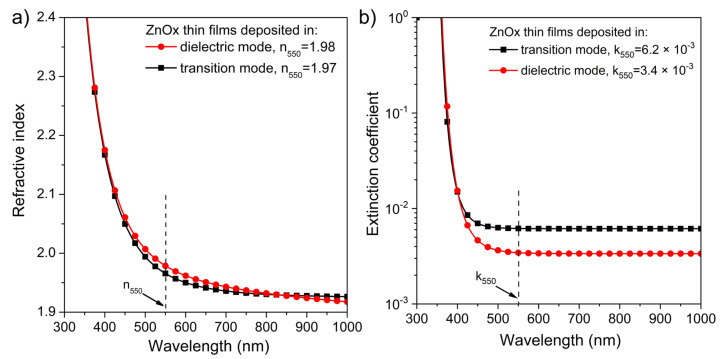
Spectral characteristics of: (**a**) refractive index and (**b**) extinction coefficient of transparent ZnOx thin films.

**Figure 10 materials-15-06551-f010:**
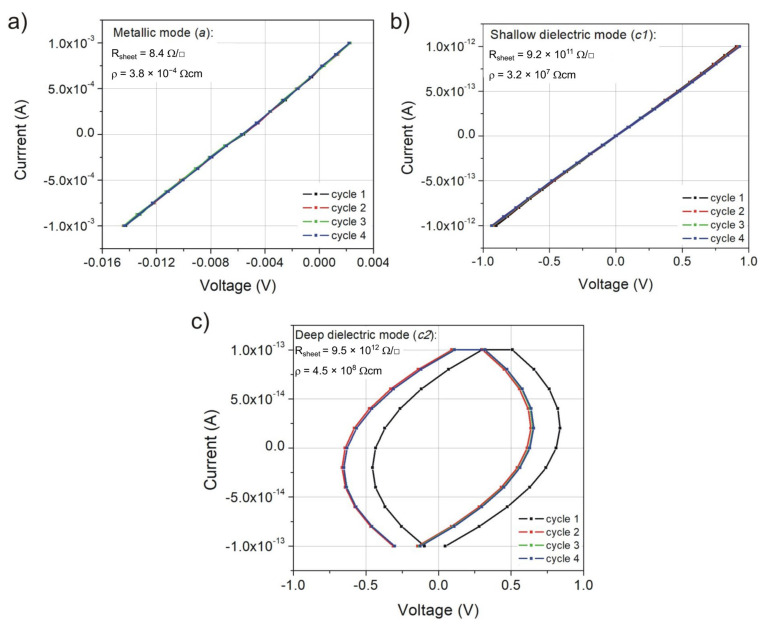
Current-voltage characteristics for ZnOx thin films deposited at modes: (**a**) metallic (*a*), (**b**) shallow dielectric (*c*1) and (**c**) deep dielectric (*c*2).

**Figure 11 materials-15-06551-f011:**
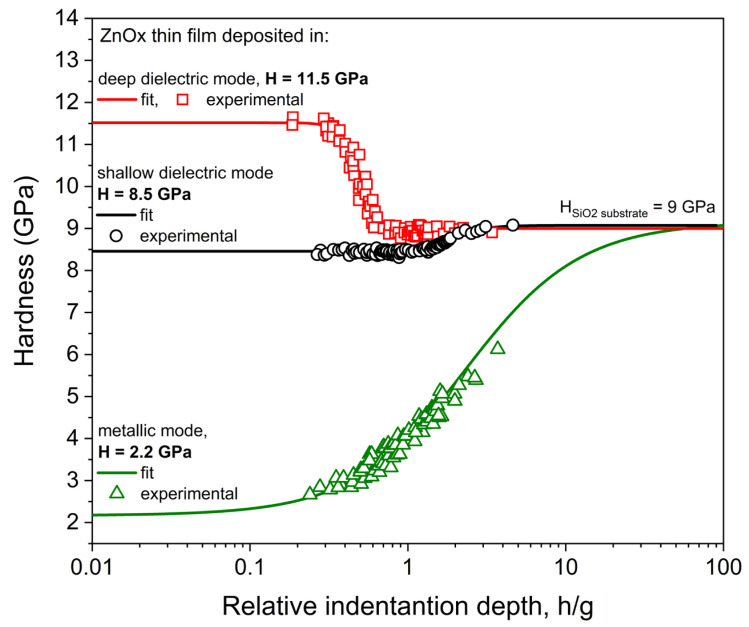
Results of hardness investigations of ZnOx thin films deposited with various technological parameters.

**Figure 12 materials-15-06551-f012:**
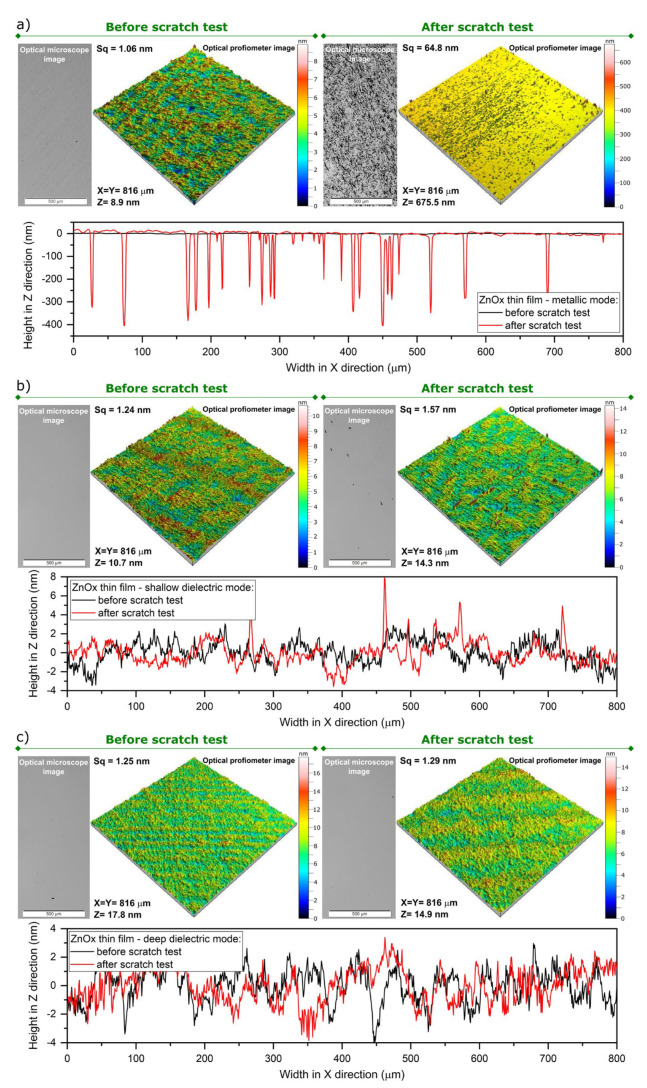
Optical microscope and profilometer images and surface cross-sectional profiles of ZnOx thin films deposited in various sputtering modes: (**a**) metallic, (**b**) shallow dielectric and (**c**) deep dielectric before and after performing abrasion tests.

**Table 1 materials-15-06551-t001:** Summary of XRD investigations of ZnOx thin films.

ZnOx Thin Film—Deposition Mode	Phase	(hkl)	2θ (o)	dhkl (nm)	dPDF (nm)	D (nm)
metallic	hexagonal-Zn	(101)	43.19	0.20927	0.20915	9.1
	hexagonal-ZnO	(002)	34.43	0.26028	0.26035	6.0
shallow dielectric	hexagonal-ZnO	(002)	34.42	0.26404	0.26035	8.7
deep dielectric	hexagonal-ZnO	(002)	34.42	0.26409	0.26035	8.1

Designations: (hkl)—set of planes, d_hkl_—measured interplanar distance, d_PDF_—standard interplanar distance from JCPDS card, D—crystallite size calculated according to the Scherrer’s equation.

**Table 2 materials-15-06551-t002:** Summary of the results of investigations for ZnOx thin films.

ZnOx Thin Film—Deposition Mode	T_λ_ (%)	λ_cut-off_ (nm)	n_550_	k_550_	R_sheet_ (Ω/Υ)	ρ (Ωcm)	Hardness (Gpa)	SRes (%)
metallic	0	0	-	-	8.4	3.8 × 10^−4^	2.2	1.64
shallow dielectric	84	361	1.97	6.2 × 10^−3^	9.2 × 10^11^	3.2 × 10^7^	8.5	78.9
deep dielectric	84	370	1.98	3.4 × 10^−3^	9.5 × 10^12^	4.5 × 10^8^	11.5	96.9

Designations: Tλ—transmittance, λcut-off—fundamental absorption edge, n550—refractive index at λ = 550 nm, k550—extinction coefficient at λ = 550 nm, R—resistance, ρ—resistivity, Sres—scratch resistance according to (Equation (1)).

## Data Availability

The data presented in this study are available on request from the corresponding author via e-mail.

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
