# Peer review of "Investigation of the Microstructure, Optical, Electrical and Nanomechanical Properties of ZnOx Thin Films Deposited by Magnetron Sputtering"

_materials, 2022, doi:10.3390/ma15196551_

Round 1

Reviewer 1 Report

Dear Authors,

Thank you for the interesting manuscript with sound experimental design.

Even though target poisoning is a well known phenomenon, and lots of research has been already performed in regard to the ZnO deposition in different modes of magnetron sputtering, the manuscript is well written and the research results obtained. I think that the manuscript is worthy to be published as is. 

Author Response

Answers to the report of Reviewer #1

on the manuscript entitled: “Investigation of the influence of technological parameters on the properties of the nanostructured ZnOx thin films deposited by pulsed reactive magnetron sputtering”

Authors: Michał Mazur, Agata Obstarczyk, Witold Posadowski, Jarosław Domaradzki, Szymon Kiełczawa, Artur Wiatrowski, Damian Wojcieszak, Małgorzata Kalisz, Marcin Grobelny, Jan Szmidt

Authors:

We would like to express our gratitude for your remarks, which let us improve our manuscript. We have taken them into account in the revised version of our paper.

Answering to the Reviewer’s remarks, we have introduced some revisions in the manuscript.

Reviewer:

Thank you for the interesting manuscript with sound experimental design.

Even though target poisoning is a well-known phenomenon, and lots of research has been already performed in regard to the ZnO deposition in different modes of magnetron sputtering, the manuscript is well written and the research results obtained. I think that the manuscript is worthy to be published as is. 

Authors:

Authors are grateful for the comment.

Reviewer 2 Report

The authors have investigated the influence of technological parameters on the properties of the nanostructured ZnOx thin films, which were fabricated using pulsed reactive magnetron sputtering. The work is interesting. However, it needs to bring more clarity and understanding to the reader by addressing the following comments:

1) Abstract: The abstract needs to be more quantitatively written. 
2) Introduction: Discuss various dimensions of ZnOx material and the advantages of thin films over different nanostructures with a comparison from recent literature.
Here are some suggestions:

1) Vemuri SK, Khanna S, Paneliya S, Takhar V, Banerjee R, Mukhopadhyay I. Fabrication of silver nanodome embedded zinc oxide nanorods for enhanced Raman spectroscopy. Colloids and Surfaces A: Physicochemical and Engineering Aspects. 2022 Apr 20;639:128336. 

2) Wang FH, Chen MS, Liu HW, Kang TK. Effect of rapid thermal annealing time on ZnO: F thin films deposited by radio frequency magnetron sputtering for solar cell applications. Applied Physics A. 2022 Mar;128(3):1-8.

3) Wójcicka A, Fogarassy Z, Rácz A, Kravchuk T, Sobczak G, Borysiewicz MA. Multifactorial investigations of the deposition process–Material property relationships of ZnO: Al thin films deposited by magnetron sputtering in pulsed DC, DC and RF modes using different targets for low resistance highly transparent films on unheated substrates. Vacuum. 2022 Sep 1;203:111299.

Result and Discussion: The section explores the different electrical, mechanical and morphological studies of as-prepared samples. However, the authors need to add EDS mapping in FESEM along with Raman Spectroscopy to bring more clarity. Also, measure the thickness of the deposited film in the FESEM image using inbuilt software. 
2) Kindly add absorption spectra for all the samples to understand if there is any red or blue shift in different samples.

Conclusion: The conclusion is well written and can be modified quantitatively. 

The manuscript consists of different grammatical errors. Kindly address them in the revised version.

Author Response

Answers to the report of Reviewer #1

on the manuscript entitled: “Investigation of the influence of technological parameters on the properties of the nanostructured ZnOx thin films deposited by pulsed reactive magnetron sputtering”

Authors: Michał Mazur, Agata Obstarczyk, Witold Posadowski, Jarosław Domaradzki, Szymon Kiełczawa, Artur Wiatrowski, Damian Wojcieszak, Małgorzata Kalisz, Marcin Grobelny, Jan Szmidt

Authors:

We would like to express our gratitude for your remarks, which let us improve our manuscript. We have taken them into account in the revised version of our paper.

Answering to the Reviewer’s remarks, we have introduced some revisions in the manuscript.

The figures can be seen in the .doc file.

Reviewer:

Abstract: The abstract needs to be more quantitatively written. 

Authors:

Abstract was corrected as follows:

“The paper presents the results of an investigation of the influence of technological parameters on the microstructure, optical, electrical and nanomechanical properties of zinc oxide coatings prepared using pulsed reactive magnetron sputtering method. Three sets of ZnOx thin films were deposited in metallic, shallow dielectric, and deep dielectric sputtering modes. Structural investigations showed that thin films deposited in the metallic mode were nanocrystalline with mixed hexagonal phases of metallic zinc and zinc oxide with crystallite size of 9.1 and 6.0 nm, respectively. On the contrary, the coatings deposited in both dielectric modes had a nanocrystalline ZnO structure with an average crystallite size smaller than 10 nm. Moreover, coatings deposited in the dielectric modes had average transmission of 84% in the visible wavelength range, while thin films deposited in the metallic mode were opaque. Measurements of electrical properties revealed that the resistivity of as-deposited thin films was in the range of 10-4 Ωcm to 108 Ωcm. Coatings deposited in the metallic mode had the lowest hardness of 2.2 GPa and the worst scratch resistance among all sputtered coatings, whereas the best mechanical properties were obtained for the film sputtered in the deep dielectric mode. Obtained hardness of 11.5 GPa is one of the highest reported up to date in the literature for undoped ZnO.”

Reviewer:

Introduction: Discuss various dimensions of ZnOx material and the advantages of thin films over different nanostructures with a comparison from recent literature.
Here are some suggestions:

1) Vemuri SK, Khanna S, Paneliya S, Takhar V, Banerjee R, Mukhopadhyay I. Fabrication of silver nanodome embedded zinc oxide nanorods for enhanced Raman spectroscopy. Colloids and Surfaces A: Physicochemical and Engineering Aspects. 2022 Apr 20;639:128336.

2) Wang FH, Chen MS, Liu HW, Kang TK. Effect of rapid thermal annealing time on ZnO: F thin films deposited by radio frequency magnetron sputtering for solar cell applications. Applied Physics A. 2022 Mar;128(3):1-8.

3) Wójcicka A, Fogarassy Z, Rácz A, Kravchuk T, Sobczak G, Borysiewicz MA. Multifactorial investigations of the deposition process–Material property relationships of ZnO: Al thin films deposited by magnetron sputtering in pulsed DC, DC and RF modes using different targets for low resistance highly transparent films on unheated substrates. Vacuum. 2022 Sep 1;203:111299.

Authors:

Authors introduced each of the mentioned literature and few more in the introduction part together with a proper comment (underlined in the reviewed manuscript).

  1. Vemuri, S.K; Khanna, S.; Paneliya, S.; Takhar, V.; Banerjee, R.; Mukhopadhyay, I. Fabrication of silver nanodome embedded zinc oxide nanorods for enhanced Raman spectroscopy, Colloids and Surf. A 2022, 639, 128336. Doi: 10.1016/j.colsurfa.2022.128336.
  2. Wang, F.H.; Chen, M.S.; Liu, H.W.; Kang, T.K. Effect of rapid thermal annealing time on ZnO:F thin films deposited by radio frequency magnetron sputtering for solar cell applications, Appl. Phys. A 2022, 128(3), 1-8. Doi: 10.1007/s00339-022-05376-5.
  3. Wójcicka, A.; Fogarassy, Z.; Rácz, A.; Kravchuk, T.; Sobczak, G.; Borysiewicz, M.A. Multifactorial investigations of the depo-sition process – Material property relationships of ZnO:Al thin films deposited by magnetron sputtering in pulsed DC, DC and RF modes using different targets for low resistance highly transparent films on unheated substrates, Vacuum 2022, 203, 111299. Doi: 10.1016/j.vacuum.2022.111299.

  1. Hussain, S.; Liu, T.; Kashif, M.; Miao, B.; He, J.; Zeng, W.; Zhang, Y.; Hasim, U.; Fusheng, P. Surfactant dependent growth of twinned ZnO nanodisks, Materials Letters. 2014, 118, 165-168. Doi: 10.1016/j.matlet.2013.12.068
  2. Kim, J.H.; Do, K.M.; Kim, J.W.; Jung, J.C.; Lee, J.H.; Moon, B.M.; Koo, S.M. Fabrication and characterization of nanocrystal-line ZnO film-based heterojunction diodes on 4H-SiC, J. Nanoelectr. and Optoelectr. 2012, 7(3), 271-274. Doi: 10.1166/jno.2012.1298
  3. Varma, T.; Sharma, S.; Periasamy, C.; Boolchandani, D. Performance analysis of Pt/ZnO schottky photodiode using ATLAS, J. Nanoelectr. and Optoelectr. 2015, 10(6), 761-765(5). Doi: 10.1166/jno.2015.1836.
  4. Lee, B.S.; Joo, Y.H.; Kim, C.I. The optical and electrical characteristics for ZnO/Al/ZnO multilayer films deposited by RF sputtering, J. Nanoelectr. and Optoelectr. 2015, 10(3), 402-407. Doi: 10.1166/jno.2015.1758.

Reviewer:

Result and Discussion: The section explores the different electrical, mechanical and morphological studies of as-prepared samples. However, the authors need to add EDS mapping in FESEM along with Raman Spectroscopy to bring more clarity. Also, measure the thickness of the deposited film in the FESEM image using inbuilt software. 

Authors:

According to the Reviewer suggestion:

  • Authors added the EDS mapping results to the manuscript as the Fig. 7.

Figure 7. EDS spectra with elemental distribution maps of ZnOx thin films deposited in metallic and deep dielectric modes

Moreover, Authors also added the EDS spectra to show that there are no peaks related to other materials showing high purity of the deposited thin films. These measurements were performed for two samples, i.e. for these deposited in metallic and deep dielectric modes. The spectrum shows peaks related to zinc and oxygen related to the thin film material composition and silicon related to the substrate material (500 mm thick Si) – small signal for metallic thin film and even smaller for sample from deep dielectric mode. This is due to the penetration depth in the EDS, which is dependent mostly upon the accelerating voltage used during measurement (and material being measured). Basically, the higher accelerating voltage the bigger penetration depth is. For example, taking just rough estimation, using 5kV e-beam the penetration depth reaches ca. 0.5 micron, while for 20 kV it can even reach 2 microns. It is clearly seen that in the case of the deep dielectric mode the peak related to the oxygen is much higher compared to the sample from metallic mode.

The rough estimation of the material composition showed the Zn:O atomic ratio near the 1:1 for the thin films from dielectric mode and almost 2:1 for the film deposited in metallic mode. This is in accordance with the predictions taking into consideration the nature of both deposition processes. However, Authors do not want to put the quantitative results into the manuscript. The EDS measurements are usually a good way of assessing the composition of the material, but studies of oxide thin films are difficult and often subject to systematic errors. This is caused by the fact that, with a reduction of the atomic number, it is increasingly difficult to ionize an atom, resulting in a weaker signal from light elements. Therefore, the measurements of the intensity of X-rays of light elements (such as e.g. oxygen) and further analysis of the composition are hard.

We think that it would be sufficient to convince readers about the qualitative material composition of zinc oxide thin films, as well as their purity since no other peaks were detected related to any other elements. Authors extended their manuscript with following comments:

  • Materials and methods section:

“The surface morphology and cross-section of as-prepared coatings were investigated using a FEI Helios Xe-PFIB field-emission scanning electron microscope (FE-SEM) equipped with EDAX energy dispersive spectroscopy (EDS) detector.”

  • Results and discussion section:

“Secondary electron images, elemental distribution maps of zinc and oxygen and EDS spectra taken for ZnOx thin films deposited in metallic and deep dielectric modes are shown in Fig. 7. The gathered maps showed homogenous distribution of each element, i.e. Zn and O. EDS spectra showed peak lines related to Zn (La line at 1.01 keV) and O (Ka line at 0.53 keV) from the thin film and Si from substrate material (Ka line at 1.74 keV). It is clearly seen that for the deep dielectric mode, the oxygen-related peak is much higher compared to the thin film from the metallic mode, which is understandable given the nature of both modes of the magnetron sputtering process. Furthermore, there were no other peaks that could come from other elements, which is evidence of the good purity of the deposited zinc oxide thin films.”

  • The thickness of the deposited films were measured using FESEM inbuilt software and shown in Fig. 6.

  • Raman spectroscopy is a non-destructive chemical analysis technique that is used for identification of chemical components in a sample. Analysis can be done to inorganic and organic samples, excluding metals or alloys. In the case of performed studies the GIXRD measurements were done in order to evaluate the structural properties of the deposited thin films - not only to determine the crystalline phases of the sputtered thin films, but also the crystallite size, which is quite important in the case of e.g. nanomechanical properties. The latter, i.e. determination of the crystallite size, is not possible with Raman spectroscopy. Moreover, one of the samples is the composite of zinc oxide and metallic zinc (which probably would not be determined by Raman spectroscopy). Therefore, Authors think that GIXRD measurements gave more insight into the microstructure of the deposited thin films than the Raman spectroscopy.

The microstructure was studied by X-ray diffraction in grazing incidence mode (GIXRD) using a Philips X'Pert Pro diffractometer equipped with the Cu Kα X-ray source with a wavelength of 1.5406 Å. Crystallite size analysis was performed according to the Debye-Scherrer equation [48] with the aid of MDI JADE 5.0 software.

“XRD analysis was performed to determine the structural properties of the prepared ZnOx thin films (Fig. 5). It was found that the diffraction pattern obtained for coatings deposited at the (a) work point (metallic mode) revealed the coexistence of metallic and oxide hexagonal phases of Zn and ZnO. The deposited thin film was nanocrystalline with small crystallites with size equal to 9.1 and 6.0 nm for Zn and ZnO, respectively. Furthermore, the zinc oxide coatings deposited at working points (c1) and (c2) had the polycrystalline structure of the hexagonal ZnO phase with the most dominant peak occurring at 2θ = 34° and associated with the (002) lattice plane. The lattice planes observed for the ZnO film deposited in the deep dielectric mode (c2 work point) were more intense than for the film deposited in the shallow dielectric mode (c1 point). In the case of ZnO films deposited at the working points (c1) and (c2), the crystallite size was equal to 7.4 nm and 6.7 nm, respectively. XRD measurements performed for thin films sputtered in dielectric modes revealed a shift of the diffraction peaks related to the ZnO hexagonal phase towards a lower angle (2θ), which may indicate the presence of a tensile stress occurring in the structure. In turn, in the case of coatings sputtered in the metallic mode, there was a visible shift of the diffraction peaks related to the metallic Zn-hexagonal phase toward a higher angle, which can testify about the presence of a compression stress. The results of the XRD measurements are summarized in Table 1.”

Reviewer:

Kindly add absorption spectra for all the samples to understand if there is any red or blue shift in different samples.

Authors:

In Fig. 1 Reviewer can see the absorption spectra for ZnO thin films deposited in the deep and shallow dielectric modes. The fundamental absorption edge is quite similar for both samples varying of only 0.03 eV.

Authors decided to add the Tauc plots with determination of the direct optical band gap energy of ZnO thin films deposited in deep and shallow dielectric mode. the optical band gap energy (Egopt) of the ZnOx thin films was calculated using the Tauc’s plot [51]. The optical band gap energy can be estimated by extrapolating the linear portion of the curves. In the case of ZnO films deposited in shallow dielectric and deep dielectric modes, the value of Egopt was equal to 3.30 eV and 3.25 eV, respectively.

The manuscript was corrected as follows:

Fig. 1. Absorption spectra of the ZnO thin films deposited in shallow and deep dielectric modes

“The optical properties of the deposited ZnOx thin films were determined based on measurements of light transmission in the wavelength range of 300-1000 nm (Fig. 7a). It was found that coatings sputtered in the metallic mode were opaque in the measured wavelength range, although their microstructure consisted of both metallic Zn and oxide ZnO phases. In turn, the ZnOx films deposited in the shallow dielectric (c1) and deep dielectric (c2) modes were well transparent with an average transparency of approximately 84 %. The results of the determination of the position of the fundamental absorption edge are also presented in Fig. 7a. The lowest cut-off wavelength (lcut-off)was equal to 361 nm for the ZnOx film deposited in the shallow dielectric mode (c1 work point). The change in the deposition mode to c2 work point resulted in a slight shift of the fundamental absorption edge towards a longer wavelength of 370 nm. Moreover, the optical band gap energy (Egopt) of the ZnOx thin films was calculated using the Tauc plot [51]. The optical band gap energy was estimated by extrapolating the linear portion of the curves presented in Fig. 7b. In the case of ZnO films deposited in shallow dielectric and deep dielectric modes, the value of Egopt was equal to 3.30 eV and 3.25 eV, respectively.”

Figure 7. Results of: a) transmission spectra and b) determination of optical band gap energy of ZnO thin films

Reviewer:

The conclusion is well written and can be modified quantitatively. 

Authors:

Conclusions have been changed as follows:

„The ZnOx thin films were deposited with different oxygen concentration in the sputtering atmosphere, which determined three different sputtering modes: metallic, shallow, and deep dielectric modes. All prepared ZnOx films were obtained with a relatively high deposition rate, which was related to the high target power density during the sputtering process, which was not found in the other works. Microstructure studies showed that thin films deposited in the metallic mode had mixed metallic (Zn) and oxide (ZnO) hexagonal crystal phases with crystallite size of 9.1 and 6.0 nm, respectively. Moreover, thin films deposited in shallow and deep dielectric modes had a polycrystalline structure of the hexagonal ZnO phase with crystallite size of 8.1 to 8.7 nm. It was also found that coatings deposited in the metallic mode were opaque, which was related to the occurrence of metallic Zn in the deposited thin film. On the contrary, thin films deposited at shallow and dielectric modes were highly transparent, with an average transparency in the visible wavelength range above 84%. On the basis of the extinction coefficient values, it can be concluded that the optical losses are almost twice smaller for ZnOx coatings deposited in the deep dielectric mode compared to films deposited at shallow dielectric mode. Measurements of electrical properties revealed that the resistivity of as-deposited thin films was dependent on the sputtering mode. For coatings deposited in the metallic mode, the resistivity was equal to 3.8·10-4 Ωcm, while for films deposited in the dielectric modes, it was significantly higher and was in the range of 107÷108  Ωcm. Measurements of nanomechanical properties revealed that metallic thin films were rather soft with hardness of 2.2 GPa, but it is worth mentioning that coatings from the deep dielectric mode had one of the highest values of hardness ever reported, i.e. 11.5 GPa. Moreover, coatings deposited in the dielectric modes were scratch resistant and the best abrasive properties were obtained for the ZnOx thin film prepared in deep dielectric mode.”

Reviewer:

The manuscript consists of different grammatical errors. Kindly address them in the revised version.

Authors:

In the revised version, the manuscript has been corrected and the changes are underlined.

Reviewer 3 Report

The manuscript Investigation of the influence of technological parameters on the properties of the nanostructured ZnOx thin films deposited by pulsed reactive magnetron sputtering presenting about ZnO materials for generalized properties.

The title of the manuscript is too long. Make it more concise.

The title must reflect specific property "on the properties of “ what properties?

In the abstract, give a sentence about key issue and the background then start on the study.

More literature about ZnO and thin film could be added in the introduction part as diverse applications

[x] Surfactant dependent growth of twinned ZnO nanodisks. Materials Letters, 2014, 118, 165-168. https://doi.org/10.1016/j.matlet.2013.12.068

[x] Fabrication and Characterization of Nanocrystalline ZnO Film-Based Heterojunction Diodes on 4H-SiC, Journal of Nanoelectronics and Optoelectronics, Volume 7, Number 3, June 2012, pp. 271-274(4)  https://doi.org/10.1166/jno.2012.1298

[x] Performance Analysis of Pt/ZnO Schottky Photodiode Using ATLAS, Journal of Nanoelectronics and Optoelectronics, Volume 10, Number 6, December 2015, pp. 761-765(5) https://doi.org/10.1166/jno.2015.1836

[x] The Optical and Electrical Characteristics for ZnO/Al/ZnO Multilayer Films Deposited by RF Sputtering, Journal of Nanoelectronics and Optoelectronics, Volume 10, Number 3, June 2015, pp. 402-407(6) https://doi.org/10.1166/jno.2015.1758

In Figure 5, why there is huge intensity difference? The intensity of the the metallic mode is more closer to amorphous phase in the low detection range.

In Shallow and deep mode, the gap is too small. It might be added in original intensity axis. 

Figure 5 and 9 should be redrawn, the resolution is very low.

Conclusions should be more brief. 

Author Response

Answers to the report of Reviewer #1

on the manuscript entitled: “Investigation of the influence of technological parameters on the properties of the nanostructured ZnOx thin films deposited by pulsed reactive magnetron sputtering”

Authors: Michał Mazur, Agata Obstarczyk, Witold Posadowski, Jarosław Domaradzki, Szymon Kiełczawa, Artur Wiatrowski, Damian Wojcieszak, Małgorzata Kalisz, Marcin Grobelny, Jan Szmidt

Authors:

We would like to express our gratitude for your remarks, which let us improve our manuscript. We have taken them into account in the revised version of our paper.

Answering to the Reviewer’s remarks, we have introduced some revisions in the manuscript.

The figures can be seen in the .doc file.

Reviewer:

The title of the manuscript is too long. Make it more concise. The title must reflect specific property "on the properties of “ what properties?

Authors:

Authors decided to change the title as follows:

“Investigation of the microstructure, optical, electrical and nanomechanical properties of ZnOx thin films deposited by magnetron sputtering”

Reviewer:

In the abstract, give a sentence about key issue and the background then start on the study.

Authors:

Abstract was corrected as follows:

“The paper presents the results of an investigation of the influence of technological parameters on the microstructure, optical, electrical and nanomechanical properties of zinc oxide coatings prepared using pulsed reactive magnetron sputtering method. Three sets of ZnOx thin films were deposited in metallic, shallow dielectric, and deep dielectric sputtering modes. Structural investigations showed that thin films deposited in the metallic mode were nanocrystalline with mixed hexagonal phases of metallic zinc and zinc oxide with crystallite size of 9.1 and 6.0 nm, respectively. On the contrary, the coatings deposited in both dielectric modes had a nanocrystalline ZnO structure with an average crystallite size smaller than 10 nm. Moreover, coatings deposited in the dielectric modes had average transmission of 84% in the visible wavelength range, while thin films deposited in the metallic mode were opaque. Measurements of electrical properties revealed that the resistivity of as-deposited thin films was in the range of 10-4 Ωcm to 108 Ωcm. Coatings deposited in the metallic mode had the lowest hardness of 2.2 GPa and the worst scratch resistance among all sputtered coatings, whereas the best mechanical properties were obtained for the film sputtered in the deep dielectric mode. Obtained hardness of 11.5 GPa is one of the highest reported up to date in the literature for undoped ZnO.”

Reviewer:

More literature about ZnO and thin film could be added in the introduction part as diverse applications

[x] Surfactant dependent growth of twinned ZnO nanodisks. Materials Letters, 2014, 118, 165-168. https://doi.org/10.1016/j.matlet.2013.12.068

[x] Fabrication and Characterization of Nanocrystalline ZnO Film-Based Heterojunction Diodes on 4H-SiC, Journal of Nanoelectronics and Optoelectronics, Volume 7, Number 3, June 2012, pp. 271-274(4)  https://doi.org/10.1166/jno.2012.1298

[x] Performance Analysis of Pt/ZnO Schottky Photodiode Using ATLAS, Journal of Nanoelectronics and Optoelectronics, Volume 10, Number 6, December 2015, pp. 761-765(5) https://doi.org/10.1166/jno.2015.1836

[x] The Optical and Electrical Characteristics for ZnO/Al/ZnO Multilayer Films Deposited by RF Sputtering, Journal of Nanoelectronics and Optoelectronics, Volume 10, Number 3, June 2015, pp. 402-407(6) https://doi.org/10.1166/jno.2015.1758

Authors:

Authors introduced each of the mentioned literature and few more in the introduction part together with a proper comment (underlined in the reviewed manuscript).

  1. Vemuri, S.K; Khanna, S.; Paneliya, S.; Takhar, V.; Banerjee, R.; Mukhopadhyay, I. Fabrication of silver nanodome embedded zinc oxide nanorods for enhanced Raman spectroscopy, Colloids and Surf. A 2022, 639, 128336. Doi: 10.1016/j.colsurfa.2022.128336.
  2. Wang, F.H.; Chen, M.S.; Liu, H.W.; Kang, T.K. Effect of rapid thermal annealing time on ZnO:F thin films deposited by radio frequency magnetron sputtering for solar cell applications, Appl. Phys. A 2022, 128(3), 1-8. Doi: 10.1007/s00339-022-05376-5.
  3. Wójcicka, A.; Fogarassy, Z.; Rácz, A.; Kravchuk, T.; Sobczak, G.; Borysiewicz, M.A. Multifactorial investigations of the depo-sition process – Material property relationships of ZnO:Al thin films deposited by magnetron sputtering in pulsed DC, DC and RF modes using different targets for low resistance highly transparent films on unheated substrates, Vacuum 2022, 203, 111299. Doi: 10.1016/j.vacuum.2022.111299.

  1. Hussain, S.; Liu, T.; Kashif, M.; Miao, B.; He, J.; Zeng, W.; Zhang, Y.; Hasim, U.; Fusheng, P. Surfactant dependent growth of twinned ZnO nanodisks, Materials Letters. 2014, 118, 165-168. Doi: 10.1016/j.matlet.2013.12.068
  2. Kim, J.H.; Do, K.M.; Kim, J.W.; Jung, J.C.; Lee, J.H.; Moon, B.M.; Koo, S.M. Fabrication and characterization of nanocrystal-line ZnO film-based heterojunction diodes on 4H-SiC, J. Nanoelectr. and Optoelectr. 2012, 7(3), 271-274. Doi: 10.1166/jno.2012.1298
  3. Varma, T.; Sharma, S.; Periasamy, C.; Boolchandani, D. Performance analysis of Pt/ZnO schottky photodiode using ATLAS, J. Nanoelectr. and Optoelectr. 2015, 10(6), 761-765(5). Doi: 10.1166/jno.2015.1836.
  4. Lee, B.S.; Joo, Y.H.; Kim, C.I. The optical and electrical characteristics for ZnO/Al/ZnO multilayer films deposited by RF sputtering, J. Nanoelectr. and Optoelectr. 2015, 10(3), 402-407. Doi: 10.1166/jno.2015.1758.

Reviewer:

In Figure 5, why there is huge intensity difference? The intensity of the metallic mode is closer to amorphous phase in the low detection range. In Shallow and deep mode, the gap is too small. It might be added in original intensity axis. 

Authors:

XRD figure was corrected. At the low 2q range there is an increase of the background, which is related to the amorphous fused silica substrates on which the ZnO thin films were deposited. For fused silica very wide peak at ca. 20-25° 2q is always present if the penetration depth of the X-ray is sufficient to reach the substrates – that is the case of thin films with thickness below 1000 nm where the substrate is also measured.

Figure 5. Diffraction patterns of ZnOx thin films deposited in metallic mode - (a) work point, shallow dielectric mode - (c1) work point and deep dielectric mode - (c2) work point

Reviewer:

Figure 5 and 9 should be redrawn, the resolution is very low.

Authors:

All figures were introduced to the manuscript in high resolution. Moreover, Authors included high resolution jpg files into submission system.

Reviewer:

Conclusions should be more brief. 

Authors:

Conclusions have been changed as follows:

“The ZnOx thin films were deposited with different oxygen concentration in the sputtering atmosphere, which determined three different sputtering modes: metallic, shallow, and deep dielectric modes. All prepared ZnOx films were obtained with a relatively high deposition rate, which was related to the high target power density during the sputtering process, which was not found in the other works. Microstructure studies showed that thin films deposited in the metallic mode had mixed metallic (Zn) and oxide (ZnO) hexagonal crystal phases with crystallite size of 9.1 and 6.0 nm, respectively. Moreover, thin films deposited in shallow and deep dielectric modes had a polycrystalline structure of the hexagonal ZnO phase with crystallite size of 8.1 to 8.7 nm. It was also found that coatings deposited in the metallic mode were opaque, which was related to the occurrence of metallic Zn in the deposited thin film. On the contrary, thin films deposited at shallow and dielectric modes were highly transparent, with an average transparency in the visible wavelength range above 84%. On the basis of the extinction coefficient values, it can be concluded that the optical losses are almost twice smaller for ZnOx coatings deposited in the deep dielectric mode compared to films deposited at shallow dielectric mode. Measurements of electrical properties revealed that the resistivity of as-deposited thin films was dependent on the sputtering mode. For coatings deposited in the metallic mode, the resistivity was equal to 3.8·10-4 Ωcm, while for films deposited in the dielectric modes, it was significantly higher and was in the range of 107÷108  Ωcm. Measurements of nanomechanical properties revealed that metallic thin films were rather soft with hardness of 2.2 GPa, but it is worth mentioning that coatings from the deep dielectric mode had one of the highest values of hardness ever reported, i.e. 11.5 GPa. Moreover, coatings deposited in the dielectric modes were scratch resistant and the best abrasive properties were obtained for the ZnOx thin film prepared in deep dielectric mode.”

Round 2

Reviewer 2 Report

The authors have mentioned adding to recent literature, EDS mapping, thickness measurement in FESEM and etc. in their response to the comments. However, the revised version of the manuscript does not contain any of the above-mentioned points. 

For example:

Authors reply,
Fig. 1. Absorption spectra of the ZnO thin films deposited in shallow and deep dielectric modes
But Figure 1 in the revised manuscript is shown as "Dependence between effective power (PE) and circulating power (PC) measured during 171 the sputtering of Zn target in Ar".

Authors Reply:

The thickness of the deposited films was measured using FESEM inbuilt software and shown in Fig. 6.

But their is no thickness measurement in the revised manuscript.

Hence, the authors are requested to modify the manuscript in accordance with their response.

Author Response

Reviewer:

The authors have mentioned adding to recent literature, EDS mapping, thickness measurement in FESEM and etc. in their response to the comments. However, the revised version of the manuscript does not contain any of the above-mentioned points.

For example:

Authors reply, Fig. 1. Absorption spectra of the ZnO thin films deposited in shallow and deep dielectric modes. But Figure 1 in the revised manuscript is shown as "Dependence between effective power (PE) and circulating power (PC) measured during 171 the sputtering of Zn target in Ar".

Authors Reply: The thickness of the deposited films was measured using FESEM inbuilt software and shown in Fig. 6. But their is no thickness measurement in the revised manuscript. Hence, the authors are requested to modify the manuscript in accordance with their response.

Authors:

Dear Reviewer – Authors have changed the manuscript according to the Reviewer’s remarks. Authors have responded to all Reviewer’s comments.

The issue is that the submission system of the MDPI does not let Authors to directly add the figures to the response in the system. Instead, they are in the attached files in the submission system:

  1. Response to the reviewer
  2. Reviewed manuscript.

Please check the manuscript file – all changes have been made according to Reviewer’s comments. EDS mapping, additional literature, SEM with thickness measurements etc. are included. Moreover, the detailed response consists of all the data required by the Reviewer.

Reviewer 3 Report

Accept in present form

Author Response

Answers to the report of Reviewer #1

on the manuscript entitled: “Investigation of the microstructure, optical, electrical and nanomechanical properties of ZnOx thin films deposited by magnetron sputtering”

Authors: Michał Mazur, Agata Obstarczyk, Witold Posadowski, Jarosław Domaradzki, Szymon Kiełczawa, Artur Wiatrowski, Damian Wojcieszak, Małgorzata Kalisz, Marcin Grobelny, Jan Szmidt

Authors:

We would like to express our gratitude for your remarks, which let us improve our manuscript.